# Characterization of the Antiproliferative Activity of *Sargassum muticum* Low and High Molecular Weight Polysaccharide Fractions

**DOI:** 10.3390/md22010016

**Published:** 2023-12-27

**Authors:** Lara Diego-González, Milena Álvarez-Viñas, Rosana Simón-Vázquez, Herminia Domínguez, Maria Dolores Torres, Noelia Flórez-Fernández

**Affiliations:** 1CINBIO, Universidade de Vigo, Grupo Inmunología, 36310 Vigo, Spain; ldiego@uvigo.gal (L.D.-G.); rosana.simon@uvigo.gal (R.S.-V.); 2CINBIO, Universidade de Vigo, Grupo Biomasa y Desarrollo Sostenible, 32004 Ourense, Spain; milalvarez@uvigo.gal (M.Á.-V.); herminia@uvigo.gal (H.D.)

**Keywords:** antitumoral, pressurized hot water, fucoidan, fractionation, alginate, rheology

## Abstract

The extract obtained by pressurized hot water extraction from *Sargassum muticum*, to recover the bioactive compound known as fucoidan, was fractionated using membranes of 100, 50, 30, 10, and 5 kDa, obtaining five retentates and the final permeate. These fractions were characterized for phloroglucinol content, protein content, sulfate content, and trolox equivalent antioxidant capacity (TEAC); apart from oligosaccharides, FTIR and molar mass distribution were also evaluated. Retentates of 100 and 50 kDa showed higher values for phloroglucinol, TEAC, and sulfate content. The rheology of the alginate fraction was also evaluated. Regarding the potential antitumoral activity, all fractions were assessed in MCF-7 cells using a metabolic activity assay based on the reduction of a tetrazolium compound, the most efficient being R100 and R50. Based on the results, these fractions were compared with commercial fucoidans at the same concentrations, and similar results were found. In addition, synergistic cytotoxic effects using two drugs commonly used in breast cancer, cis-Platinum (cis-Pt) and 5-fluorouracil (5-FU), were tested in combination with R100 and R50. Promising results were obtained when the retentate and the drugs were mixed, showing an improvement in the cytotoxicity induced by the chemotherapy.

## 1. Introduction

The brown seaweed *Sargassum muticum* (order Fucales) is an invasive seaweed of the Atlantic coast (origin Japan). It has been used to extract compounds of industrial interest to apply in nutricosmetics, food, cosmetics, or biomedicine to revalorize this by-product.

Several studies have confirmed that seaweeds generally contain bioactive compounds, such as polysaccharides, proteins, lipids, pigments, and polyphenols, with potential activity in the biomedical field due to their antibacterial, antifungal, antioxidant, antitumor, antiviral, and other properties [1]. The main component of seaweeds is polysaccharides, a biopolymer comprised of carbohydrate monomers linked by glycosidic bonds. In particular, the main polysaccharide in brown algae is fucoidan, which is usually composed of fucose and sulfate groups but can also contain galactose, mannose, xylose, glucose, and others (Figure 1). Furthermore, the chemical structure and the position of the sulfate groups are related to the biological properties of this polymer [2,3,4]. Brown algae also contain other polysaccharides in low concentrations, such as alginic acid and laminarin [5,6].

The composition of the seaweeds is related to several factors, such as the season in which the raw material is collected or abiotic factors, among others [7]. To obtain bioactive compounds, different extraction technologies have been used: ultrasound-assisted extraction, microwave-assisted extraction, supercritical fluid extraction, pressurized hot water extraction, or enzyme extraction, which allow an environmentally friendly extraction to obtain high-value compounds from algae. The operating conditions selected play a key role in the extraction of these bioactive compounds [4,8,9,10].

Hydrothermal extraction with pressurized hot water has been used as a green extraction technology to recover bioactive compounds from seaweeds [11]. In a previous study, *S. muticum* was used as a raw material to extract bioactive compounds at different temperatures to find the most suitable extraction temperature to obtain maximum fucoidan content [12]. *Laminaria japonica*, another brown seaweed, was also used as a raw material to extract the polysaccharide fraction from different extraction methods to make a comparison between the results, hot water under pressurized conditions being one of them [13]. Phlorotannin-rich fractions have been obtained by other authors from other brown seaweeds using the same extraction technologies [14]. On the other hand, membrane fractionation allows for the preparation of several fractions of different molecular weights. Moreover, these fractions can have different compositions and, therefore, different biological properties and behaviors [15].

The main aim of this work was the fractionation and purification of the extract obtained by pressurized water extraction at 170 °C from the brown seaweed *S. muticum* by membrane processing to obtain purified fractions using seawater as an extraction solvent.

The effect on the cell viability of the fractions was evaluated in the breast cancer cell line MCF-7 and compared with commercial fucoidans (from *Macrocystis pyrifera* and *Undaria pinnatifida*).

Breast cancer continues to be one of the most prevalent tumors and the first cause of death by cancer in women, despite the latest advances in diagnosis and treatment [16]. Hence, there is an urgent need to find more suitable therapies with low toxicity. The use of natural compounds with antiproliferative activity can be a safe and cost-effective alternative to improve the limited therapeutic effects of conventional therapies, such as chemotherapy.

For that reason, we have also tested the effect of the most active fractions in combination with two of the most frequently used chemotherapeutic drugs (cis-Platinum (cis-Pt) and fluorouracil (5-FU)).

## 2. Results and Discussion

### 2.1. Extraction Process and Membrane Fractionation

The parameters used in the extraction process using *S. muticum* as a raw material to obtain a bioactive fraction rich in fucoidan were based on previous work [12]. In the current work, the main variable incorporated was the use of seawater as a solvent to assess its influence on the fractional recoveries. After extraction, the free alginate liquid fraction was obtained and fractionated by membrane process using different cut-offs, from 100 kDa to 5 kDa, to study the properties, behavior, and activities (Figure 2).

### 2.2. Characterization of the Solid Fraction

The pressed alga *S. muticum* and the solid residue obtained after extraction were analyzed to determine the proximal composition of both. Moisture, ash, sulfate, protein, acid-insoluble residue, and carbohydrate content were determined; minerals and heavy metals results are summarized in Table 1. The moisture content of the raw material was reduced from 85.6% to 67.9% by the pressing process. Therefore, this pressing effect allows for the removal of almost 20% of the water present in *S. muticum*; a viscous liquid was also removed. Probably, this liquid could be part of the alginate fraction due to the observed viscosity level. Similar results were obtained for both algae and pressed algae, except for the sulfate content, which showed a low value compared to the original *S. muticum*. The values obtained were compared with other works that found similar results [17,18]. Also, in the case of the pressed algae, an effect on the carbohydrate content was observed; possibly the pressing process allowed the removal of the major fraction of the alginate, improving access to the cell wall and consequently the carbohydrate content. The value obtained for the sulfate content in the solid residue could be due to sulfate being moved to the liquid phase, obtaining a rich fraction of fucoidan.

Minerals and heavy metals were also evaluated and are shown in Table 1. A comparison of the results obtained for *S. muticum* in the present work and other brown seaweeds was undertaken, and comparable values were found [19,20].

### 2.3. Rheology of Alginate

Figure 3a presents the variation of the apparent viscosity with the shear rate for the alginate extracted after hydrothermal treatment at 170 °C compared with that from commercial alginate. Both biopolymers tested at the same conditions exhibited similar viscous profile behavior as reported elsewhere [21]. Two different regions were observed in the flow curves, the first exhibiting a Newtonian plateau below 10 1/s and the second one featuring shear-thinning behavior above that shear rate. At a fixed shear rate, the alginate viscosity of the extracted biopolymer was higher (about 10-fold) than that of its commercial counterpart. The tested alginate using seawater as an extraction solvent featured higher values of apparent viscosity than those reported for the alginate extracted after hydrothermal treatment at 170 °C in the presence of distilled water [22]. It should be noted that no hysteresis phenomena were identified in both tested alginates.

The tan δ (viscous modulus/elastic modulus, G″/G′) obtained from the corresponding frequency sweeps is presented in Figure 3b. Liquid-like behavior is observed for both tested alginates without notable differences in the magnitude of this parameter. The observed profiles indicated that the viscous behavior predominated over the elastic one over the tested frequency range. A high dependence of the tan δ on the frequency was also identified. The achieved magnitudes are slightly higher than those found for the biopolymer after hydrothermal extraction at the same temperature in the presence of distilled water [22]. In this case, hysteresis loops were not observed for any of the two tested alginates.

### 2.4. Characterization of the Liquid Fraction

The free alginate liquid fraction and several fractions were analyzed and summarized in Table 2. The effect of the membrane cut-off was observed in all the parameters characterized. Phloroglucinol content, TEAC value, sulfate content, and protein content were maximum in the R100 fraction, finding the minimum content on conductivity. According to these results, the most active fraction could be R100. Yoo and coauthors (2019) found that the high molecular weight (HMW) fraction was an effective immunostimulant in immunosuppressive conditions [23]. Other work studied the cytotoxic effects of low molecular weight (LMW) and HMW fractions of several fucoidans in two breast cancer cells. The results obtained from some samples of fucoidan from *U. pinnatifida* by Lu et al. (2018) [24] suggested that the LMW was more active than the HMW fraction, observing a dose-dependent cell inhibition in the presence of the LMW [24]. The values of protein and sulfate content are in line with those obtained for the same sample, *S. muticum,* using similar experimental conditions to obtain several fractions by membrane cut-off [15].

Biological activity can be found in brown seaweed, where the highest amounts of sulfate and fucose (fucoidan) are present. It is observed in Figure 4 that the greatest carbohydrate recovery was found in R100, which has the highest content of fucose. Similar results were found for several types of fucoidan, where the fucose content was typically greater than that of other saccharides [24]. Other authors have also obtained analogous results, where fucose was the primary saccharide [23]. Birgersson and colleagues (2023) conducted the polysaccharide fractionation of two brown seaweeds: *Saccharina latissima* and *Alaria esculenta*. The study showed that *S. latissima* had a sugar content of around 30% [25]. Another study was carried out on *Fucus vesiculosus* seaweed, and the results indicated that the fucoidan content was higher in the largest fraction, demonstrating similar findings to the present study [26]. The membrane fractionation process had a discernible impact on the proximal composition, as evidenced by this experiment.

The molar mass distribution profile was assessed by size-exclusion chromatography. Figure 5 reveals the membrane fractions with varying distributions. Specifically, in the case of the R100, the molar mass values ranged from 23.6 to 277 kDa, while the other fractions were above 277 kDa. Lu et al. (2018) analyzed several fucoidans of different origins, some of which showed profiles with low molecular weight, while others had values above 440 kDa and <2000 kDa [24]. Other work has evaluated the molecular weight profile of fucoidan extracted from *U. pinnatifida*, revealing peak values of 258 kDa for HMW and 54 kDa for LMW [23]. The differences found in the works could be attributable to the seaweed origin and the extraction technology used.

The FTIR spectra for the fractions of fucoidan and the crude extract (L170) have been represented in Figure 6. The absorption peaks were found at 1210, 1420, and 1640 cm^−1^ associated with the absorption of asymmetric and symmetric stretching vibrations of carboxylate anions (COO−) and O-H stretching, respectively [27,28,29]. Also, a slight band at 622 cm^−1^ was observed; this peak was associated with asymmetric and symmetric O=S=O deformation of sulfates [12]. In the case of R100, a peak at 1027 cm^−1^ was observed; this band could be attributable to the C-O and C-C stretching vibrations of the pyranose ring [30]. In conclusion, Figure 5 exhibited the main peaks kept in all the fractions except for R100.

### 2.5. Reduction in Tumor Cell Viability Induced by the Fractions In Vitro

The retained fractions were first plated and incubated on agar plates to confirm the absence of any bacterial or fungal contamination that could interfere with the biological tests (Appendix A).

The MTS colorimetric test was used to study the cytotoxic or antiproliferative activity of the fractions. All samples were added at different concentrations to characterize any dose-dependent effect. Figure 7A shows the changes in the cell viability of the MCF-7 cell line after 48 h of incubation. Interestingly, the intermediate concentrations (from 15.6 to 250 µg/mL) were those that induced the greatest reduction in cell viability, but only R10, R50, and R100 showed a statistically significant effect. However, at 72 h (Figure 7B), all fractions showed a significant reduction in cell viability compared to the non-treated cells, and unlike the previous time, the cytotoxic effect occurs in a dose-dependent manner except for the HMW fractions (R100 and R50). The apparent increase in cell viability observed at the highest concentrations tested, or the lack of a dose-dependent effect, could be due to the presence of antioxidant compounds. These compounds can interfere with the MTT and MTS cell viability assays [31,32]. To rule out this interference and to characterize the time-dependent reduction in cell viability of the most active fractions, R50 and R100, we used the real-time cell analyzer xCELLigence. The results showed time-dependent inhibition of MCF-7 cell proliferation with both fractions (Appendix A). However, a lack of a dose-dependent effect was also observed for R100 at the highest concentrations tested. Albeit both R50 and R100 were able to slow down the proliferation rate of the cells, only the R50 fraction at the highest concentration tested (800 µg/mL) induced a decrease in the cell viability that could be associated with a cytotoxic effect.

The antiproliferative activity of *S. muticum* in various cell lines was also confirmed with aqueous and methanolic extracts obtained by different extraction techniques [33,34]. In brief, we have shown a reduction of up to 30% in the cell viability of lung, colon, and pancreatic adenocarcinoma and a glioblastoma cell line at higher concentrations (500 mg/mL) by using a crude extract from *S. muticum* obtained by ultrasound-assisted aqueous extraction [33]. Namvar and colleagues (2013) demonstrate the cytotoxic and dose-dependent effects of an *S. muticum* polyphenol-rich methanolic extract after 24 h of incubation with two different breast cancer cell lines. The half-maximum inhibitory concentration (IC_50_) was 22 μg/mL for MCF-7 and 55 μg/mL for the MDA-MB-231 cell line [34]. In addition, they studied the cell cycle and caspase activation induced by the extract in the cells, finding that the antiproliferative activity of the polyphenol-rich *S. muticum* extract was mediated by the induction of apoptosis.

Likewise, metal oxide nanoparticles synthesized from an aqueous extract of this algae also showed a cytotoxic effect in both murine [35] and human cell lines [36] of different origins (lymphocytic, breast, colon, hepatic, etc.) after 72 h of incubation that could be partially due to the algae components, although the extract alone was not tested for comparison.

In the present work, we have compared, for the first time, the influence of the molecular weight of *S. muticum* polysaccharide fractions obtained by PHWE and membrane fractioning on the cell viability of MCF-7 cells.

Based on the results, we can conclude that R50 and R100 were more efficient than R30, R10, and R5. Interestingly, the retentates obtained in the first fractions by using membranes with a higher cut-off (100 and 50 kDa) contain the lowest molecular weight polysaccharides, as shown by size exclusion chromatography (Figure 4). Hence, our results agree with previous studies showing the superior antitumoral activity of low molecular weight fucoidans compared with their high molecular counterpart, as well as the influence of the extraction method on the activity [37].

Fucoidans have been described to modulate pathways and mechanisms that are associated with their antitumoral effect [37,38,39,40]. These sulfated polysaccharides can inhibit the Phosphatidylinositol-3-kinase (PI3K)/AKT/mammalian target of rapamycin (mTOR) and mitogen-activated protein kinases (MAPKs) signaling pathways related to cell proliferation and survival and induce the activation of caspases. In addition, they can reduce the angiogenesis and the expression of different matrix metalloproteinases (MMPs) and, consequently, the metastasis of the tumoral cells. It has also been described that fucoidans can arrest the cell cycle in the G0/G1 phase in tumoral cells and modulate the immunosuppression in the tumor environment.

According to the results obtained for the TEAC value and phloroglucinol, sulfate, and protein content (Table 2 and Figure 2), the cell viability studies demonstrated that R100 retains the highest antiproliferative activity of the *S. muticum* aqueous extract. The superior biological activity found for R50 could be due to the presence of other active compounds.

The effect on cell viability of these fractions at 72 h was compared with two different commercial fucoidans (from *M. pyrifera* and *U. pinnatifida*). R100 and R50 induced a similar or even higher effect in the MCF-7 cells when using the same concentrations, as shown in Figure 8. The fact that R50 and R100 show a similar or greater effect than that obtained with other fucoidans [24,41] demonstrates their potential as antitumoral therapeutic agents. In addition, our results support PHWE and membrane fractioning as an efficient technique to obtain biologically active compounds from brown algae.

### 2.6. Synergistic Cytotoxic Effect of R100 and R50 with Chemotherapeutic Drugs

Chemotherapeutic agents, such as cis-Pt and 5-FU, are the first-line treatment against various solid tumors [42]. However, their efficacy and use in the clinic are sometimes diminished by their side effects and the intrinsic or therapy-induced drug resistance [43,44]. For this reason, combined therapies are increasingly used in the clinic and being tested in clinical trials [45]. The use of biologically active and safe compounds that can increase the therapeutic effect of a chemotherapeutic drug while reducing the dose and associated side effects would be a desired alternative to the available therapies.

Because R100 and R50 were the fractions that showed the greatest antiproliferative activity in the MCF-7 cell line, we tested their potential synergistic cytotoxic effect in vitro with two commonly used drugs in the treatment of breast cancer, cis-Pt and 5-FU [46,47].

Two different concentrations (31 and 62 µg/mL) of the fractions were incubated for 72 h in combination with the half-maximum inhibitory concentration (IC_50_) of the drugs (Appendix A; cis-Pt: 12 µM and 5-FU: 100 µM). Both concentrations showed a synergistic cytotoxic effect in combination with cis-Pt and 5-FU in the MCF-7 cell line (Figure 9). R50 at 62.5 µg/mL in combination with cis-Pt and 5-FU at IC_50_ showed the highest inhibitory effect on cell viability because almost no viable cells were detected.

Our results agree with the work carried out by Yan et al., 2023 [48] and Huang et al., 2021 [49]. They also found that fucoidan was able to enhance the toxicity of cisplatin in human oral cancer and 5-FU in colon cell lines, respectively. In the latter work, they used a low molecular weight fraction from *S. hemiphyllum*. The therapeutic effect of fucoidan in both tumor types was mediated by several of the mechanisms described above, such as the inhibition of the PI3K/AKT pathway or cell cycle arrest. In addition, fucoidan can protect against the systemic toxicity induced by chemotherapy, such as the renal toxicity associated with the administration of cisplatin [50].

## 3. Materials and Methods

### 3.1. Raw Material

The brown seaweed *S. muticum* was collected in August 2016 at Praia a Mourisca (4.224176° N, −8.771932° W, Pontevedra, Spain). The seaweed was washed with tap water to remove sand and epiphytes; afterwards, the seaweed was stored at −18 °C in hermetic and dark plastic bags. The seaweed defrosted was cut, obtaining a particle size of around 0.5 cm. Seawater was collected at 42°27′00″ N 8°37′59″ W and stored at 4 °C for 48 h.

### 3.2. Pressing Process

*S. muticum* was pressed (Enerpac RC106, Menomonee Falls, WI, USA), and 50 g of wet seaweed was applied with a force of 2 MPa. After this process, the liquid phase and the solid phase were obtained. The brown seaweed pressed was characterized and used for the extraction process.

### 3.3. Extraction Treatment

Samples of *S. muticum*, previously pressed, were defrosted and mixed with seawater. The liquid:solid ratio used was 30:1 (*w*/*w*, d.b.), according to previous works [51]. The mixture was heated to 170 °C (non-isothermal treatment) in a stirred stainless steel reactor (model 4848, Parr Instr., St. Moline, IL, USA). When the temperature of 170 °C was achieved, the reactor was cooled to room temperature [12]. After the process, the liquid and solid phases were split by filtration and characterized.

### 3.4. Alginate Precipitation

Alginate was precipitated from the liquid phase obtained after pressurized hot water extraction at 170 °C. The precipitation was performed by adding 1% (*w*/*w*) of calcium chloride (Acros Organics, Geel, Belgium) to the extract of *S. muticum*. The mixture was stirred overnight at room temperature and then centrifugated at 4500 rpm for 40 min (Rotixa 50RS, Hettich Zentrifugen, Mülheim an der Ruhr, Kirchlengern, Germany).

### 3.5. Rheology of the Biopolymer

Aqueous dispersions of the extracted alginate were made at 2%, using commercial alginate for comparative purposes. The alginate was stirred at 1500 rpm and kept at room temperature for 1 h [22]. The rheology of the samples was analyzed on an MCR302 controlled-stress rheometer (Anton Paar, Ostfildern, Germany) utilizing a sand-blasted parallel plate geometry (1 mm gap, 25 mm diameter) at 25 °C. Paraffin oil was used to seal the edges of the samples, which were rested on the rheometer plates for 5 min before testing. Steady-state shear measurements were monitored by decreasing and subsequently increasing the shear rate following a logarithmic sweep. Small-amplitude oscillatory shear measurements were conducted at 1.5 Pa within the linear viscoelastic region. All measurements were made at least in triplicate.

### 3.6. Membrane Fractionation

The liquid phase from PHWE was fractionated using cut-off membranes from 5 to 100 kDa (Merk-Millipore, Darmstadt, Germany). The retentates were recovered, and the permeates were transferred to a next-size molecular weight cut-off membrane. The starting point was the self-hydrolysis free-alginate liquid phase obtained from *S. muticum* diluted with distilled water (1:5) and filtered through a membrane from the highest until the lower membrane cut-off. The operation condition was focused on reaching a volume concentration ratio of 5. The fractions obtained were >100 kDa, 50–100 kDa, 30–50 kDa, 10–30 kDa, 5–10 kDa, and <5 kDa, which were noted as R100, R50, R30, R10, R5, and P5, respectively, where R means retentate and P means permeate. These fractions were characterized.

### 3.7. Characterization of Raw Material and Residual Solid Phase

The moisture content was determined gravimetrically using a laboratory oven (Trade, Madrid, Spain) at 105 (±2) °C for 48 h. Ash content was also evaluated gravimetrically using a muffle oven (Carbolite, Hope Valley, UK) at 575 °C for 6 h. The protein content was determined by the Kjeldahl method, and the result was calculated using a conversion factor (5.38) [52].

Carbohydrate content was determined by high-performance liquid chromatography (HPLC); previously, a quantitative acid hydrolysis was performed. The seaweed sample was hydrolysated with 72% H_2_SO_4_ at 30 °C for 1 h in a water bath; after that, a second hydrolysis was performed, diluting at 4% H_2_SO_4_. The operation conditions were: 121 °C for 1 hour in an autoclave (P Selecta, Madrid, Spain). Immediately, the solution was filtrated, and the two phases were separated (the liquid phase and the solid phase). The solid phase was put in a laboratory oven for 48 h at 105 °C to quantify the acid-insoluble residue (AIR). The liquid phase was used to determine glucose, xylose (xyl), galactose (gal), mannose (man), and fucose in an HPLC (1100 series, Agilent, Santa Clara, CA, USA) using an Aminex HPX-87H column 300 × 7.8 mm (BioRad, Hercules, CA, USA) with a refractive index (RI), and the operation conditions were: 0.003 M H_2_SO_4_ at 50 °C with 0.6 mL/min as a flow rate.

Minerals and heavy metals were determined using different techniques: atomic emission spectrophotometry (AES), atomic absorption spectrophotometry (AAS), inductively coupled plasma mass spectrometry (ICP-MS), and cold vapor atomic absorption spectrometry (CVAAS). Sodium and potassium were analyzed by AES; on the other hand, zinc, calcium, magnesium, iron, and copper were analyzed by AAS; cadmium and lead were analyzed by ICP-MS; and mercury was evaluated by CVAAS. Previously, microwave-assisted digestion (Marsxpress, CEM) equipment was used to prepare the samples: ashes (0.3 g) of the samples were mixed with HNO_3_ (10 mL) and H_2_O_2_ (1 mL), and the operation conditions were 1600 W for 15 min; maintaining for 10 min at 200 °C.

### 3.8. Characterization of Liquid Phases

#### 3.8.1. Phloroglucinol Content

The content of phloroglucinol was determined using the method described previously in [53]. In brief, 1 mL of sample, or distilled water for the blank, was added to a mixture of 1 mL of Folin–Ciocalteu (Scharlau, Madrid, Spain) 1N solution and 2 mL of Na_2_CO_3_ 20%. The mixture was stirred in a vortex and incubated in darkness for 45 minutes at room temperature. Absorbance was measured at 730 nm against a blank in a spectrophotometer (Thermo Scientific Evolution 201 UV-Visible, Waltham, MA, USA). Phloroglucinol reagent (Aldrich, St. Louis, MO, USA) was used to prepare the calibration curve, and the results were expressed as phloroglucinol equivalents.

#### 3.8.2. Trolox Equivalent Antioxidant Capacity Assay

The method known as trolox equivalent antioxidant capacity (TEAC) was used to evaluate the antioxidant activity [54]. Briefly, 10 μL of liquid extract (or distilled water for the blank) was mixed with 1 mL of diluted ABTS^+^ solution (using PBS to dilute) and incubated at 30 °C for 6 min. Absorbance inhibition was read at 734 nm, and the results were expressed as the TEAC value. Trolox reagent (Aldrich, St. Louis, MO, USA) was used to perform the standard curve.

#### 3.8.3. Sulfate Content

The gelatin–barium chloride method was used to determine the sulfate content following the protocol described by Dodgson (1961) [55]. The BaCl_2_-gelatin reagent was prepared, and gelatin powder (Scharlau, Madrid, Spain) at 0.5% (*w*/*v*) was dissolved in hot water (70–80 °C). The solution was cooled to room temperature; after that, the gelatin was kept at 4 °C for at least 6 h. BaCl_2_ (Scharlau, Madrid, Spain) was then added at 0.5% (*w*/*v*) to give a cloudy solution, which must be kept at 4 °C for at least 6 h before use. An aliquot of 0.1 mL of the liquid phases and distilled water (as blank) were placed in test tubes and mixed with 1.9 mL of trichloroacetic acid solution (Sigma-Aldrich, Barcelona, Spain) at 4% (*w*/*v*) and 0.5 mL of gelatin-BaCl_2_ reagent, and then incubated at room temperature for 15 minutes before measurement. The absorption of the samples and the blank was read at 500 nm. The standard curve was performed with K_2_SO_4_ (Panreac, Madrid, Spain).

#### 3.8.4. Protein Content

The protein content was evaluated according to the method developed by Bradford (1976) following the supplier instructions provided by the Bradford reagent (Panreac, Berlin, Germany) [56]. Briefly, 100 µL of sample or blank were placed in a test tube, and 1 mL of Bradford reagent was added above. The mixture was stirred in a vortex and measured. The standard curve was performed using bovine serum album (BSA, Sigma, St. Louis, MO, USA), and then the results were expressed as BSA equivalents. The absorbances (Abs) of the samples were measured at 595 nm.

#### 3.8.5. Carbohydrate Content

The liquid phase obtained by non-isothermal hot pressurized water extraction from *S. muticum* and the several retentates obtained by fractionation using membrane cut-off were analyzed by high-performance liquid chromatography (HPLC). Previously, the samples were diafiltered (molecular weight cut-off, 100–500 Da, SpectrumLabs, Santa Monica, CA, USA). Glucose, rhamnose, fucose, acetic acid, and formic acid were determined using a 300 × 7.8 mm Aminex HPX-87H column (BioRad, Hercules, CA, USA) operating at 60 °C with 0.003 M H_2_SO_4_ at 0.6 mL/min as the mobile phase. The oligosaccharide content was determined from the monosaccharide concentrations in the samples after hydrolysis (4% sulfuric acid at 121 °C, 20 min).

#### 3.8.6. High-Performance Size Exclusion Chromatography

The molar mass distribution of the liquid samples was analyzed by high-performance size exclusion chromatography (HPSEC) using a column TSKGel SuperMultipore PW-H (6 × 150 mm, Tosoh Bioscience, Berlin, Germany) with a TSKGel guard column SuperMP (PW)-H (4.6 × 35 mm, Tosoh Bioscience, Germany). The mobile phase used was Milli-Q water at 0.6 mL/min, and the standards were poly(ethylene oxide) from 2.36 × 10^4^ to 7.86 × 10^5^ Da (Tosoh Bioscience, Tokio, Japan).

#### 3.8.7. Fourier Transform Infrared Spectroscopy

The lyophilized samples were mixed with potassium bromide. The FTIR spectra of the samples were recorded at 400–4000 cm^−1^ at 25 scans/min (Nicolet 6700, Thermo-Scientific, Waltham, MA, USA). The source used was IR, the detector was DTGS KBr, and the software was OMNIC. The assay was performed at least in duplicate for all samples.

#### 3.8.8. Cell lines and Antitumoral Reagents

The MCF-7 (breast) cell line was obtained from the American Type Culture Collection (ATCC) (Manassas, VA, USA) and cultured in Roswell Park Memorial Institute (RPMI) medium supplemented with 10% fetal bovine serum (FBS) and 1% antibiotics (Penicillin/Streptomycin) at 37 °C in a 5% CO_2_ atmosphere. Every two or three days, cells were diluted following the recommended protocols.

Cis-Pt, 5-FU, and the commercial fucoidans from *M. pyrifera* and *U. pinnatifida* were purchased from Sigma-Aldrich (Merck, Darmstadt, Germany).

#### 3.8.9. Cell Viability Assays

##### Stock Solutions

The stock concentrations were: R100 (11.36 mg/mL), R50 (12.98 mg/mL), R30 (8.68 mg/mL), R10 (12.27 mg/mL), R5 (7.93 mg/mL), cis-Pt (2 mM), and 5FU (40 mM). In all cases, Milli-Q water was used as a solvent.

##### MTS

The MTS Cell Proliferation Colorimetric Assay Kit (BioVision Incorporated, Waltham, MA, USA) was used to measure cell viability.

Briefly, 1 × 10^4^ cells/well (MCF-7) and 7 × 10^3^ cells/well (A549) were seeded in 96-well plates. After 24 h, the fractions or the chemotherapeutic drugs were added. Seven serial dilutions (1:2) were performed, with the maximum concentration tested being 1000 µg/mL for the *S. muticum* fractions, 100 µM for cis-Pt, and 2000 µM for 5-FU. To avoid any interference in the measurements, the samples or drugs in the culture medium and the culture medium alone were used as controls.

After 48 or 72 h of incubation, the MTS reagent was added according to the manufacturer’s instructions. Finally, the Abs was measured at 490 nm in a plate reader (EnVision, Perkin-Elmer Inc., Norwalk, CT, USA), and the percentage of cell viability was calculated following the formula:$$\%Cell\;viability=\frac{Abs\left(Cells+Sample\right)-Abs(Sample)}{Abs\left(Cells\right)-Abs(RPMI)}\times 100$$

##### xCELLigence

The xCELLigence system allows the monitoring of cell growth in real-time by recording changes in the impedance of the cells every 15 min. The values of the impedance are transformed into a cell index by the xCELLigence software V1 [57].

A total of 1 × 10^4^ cells/well of the MCF-7 cell line were seeded in E-plates (Agilent, Santa Clara, CA, USA) and placed into the xCELLigence^®^ RTCA DP Instrument (RocheDiagnostics, Penzberg, Germany). When cells achieved their exponential phase of growth, around 24 h of incubation, R50 and R100 were added at different concentrations (800, 400, 200, 100, and 50 µg/mL) and incubated for 96 h. The cell index was normalized to the time of adding the fractions, and the percentage of cell viability in the cells incubated with the fractions was calculated with respect to the untreated cells at different time points (24, 48, 72, and 96 h).

##### Statistical Analysis

The normal distribution of the samples was determined by a Shapiro–Wilk test, and a Student’s *t*-test or a Mann–Whitney test was used to find the statistically significant differences in the treatments compared to the untreated cells. The GraphPad Prism 8 software (GraphPad Software, Inc., La Jolla, CA, USA) was used for the statistical analysis and data representation.

## 4. Conclusions

The present study describes the successful extraction of purified fractions from *Sargassum muticum* using seawater as an extraction solvent. R100 showed the highest phloroglucinol, sulfate, oligosaccharide, protein content, and TEAC value. All fractions showed a reduction in cell viability on the human breast cancer cell line MCF-7, with R100 and R50 being the most active ones and the fractions that contained the lowest molecular weight polysaccharides. In addition, R100 and R50 were able to synergistically increase the cytotoxic activity of cis-Pt and 5-FU in the MCF-7 cell line. Hence, these two fractions could be beneficial in combined antitumoral therapy; however, the safety and therapeutic effect of the compounds should be evaluated in vivo.

## Figures and Tables

**Figure 1 marinedrugs-22-00016-f001:**
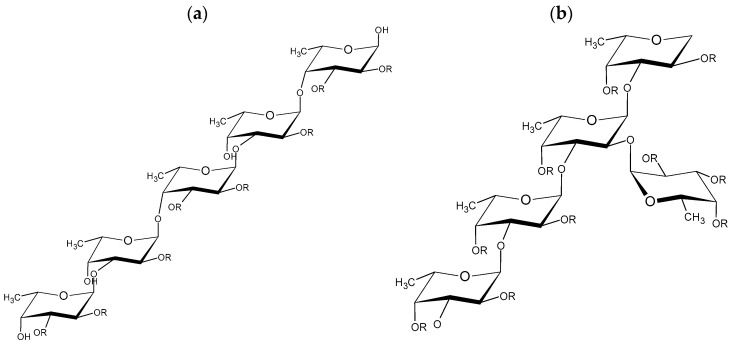
Representative models of the chemical structures of fucoidan from (**a**) *Fucus vesiculosus* and (**b**) *Saccharina latissima*. Note: R = SO_3_^−^.

**Figure 2 marinedrugs-22-00016-f002:**
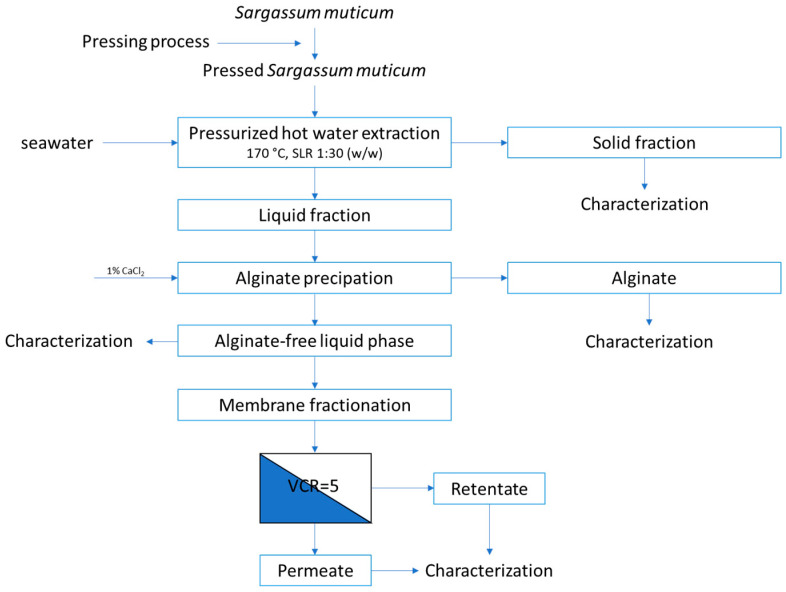
General diagram of the extraction process and membrane fractionation of the brown seaweed *S. muticum*. Note: VCR means volume concentrate ratio.

**Figure 3 marinedrugs-22-00016-f003:**
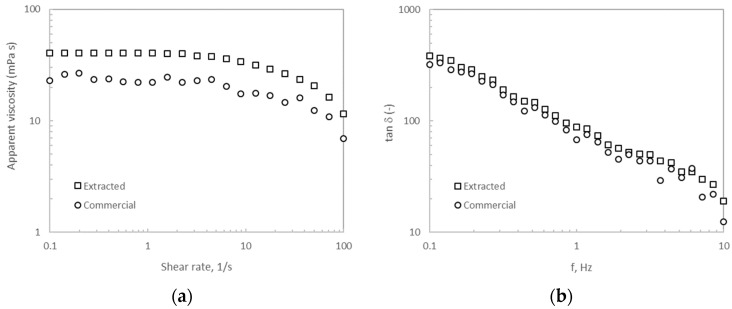
(**a**) Steady-state shear and (**b**) small-amplitude oscillatory shear measurements of the extracted alginate after treatment at 170 °C (squares) when compared with its commercial counterpart (circles).

**Figure 4 marinedrugs-22-00016-f004:**
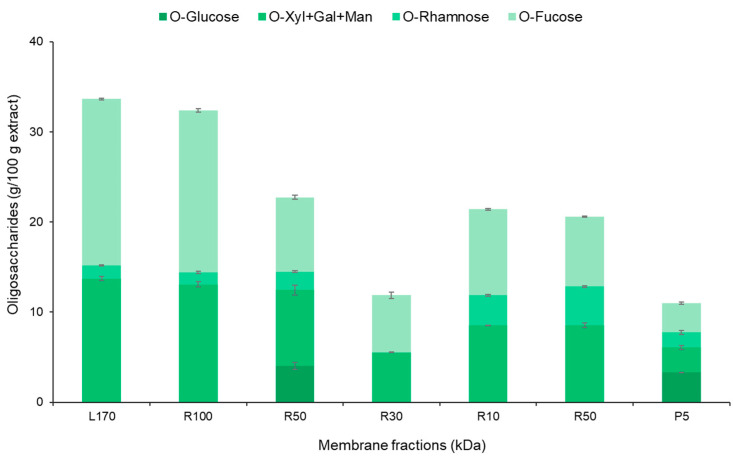
Oligosaccharides and other compounds analyzed in the initial liquid phase and the fractions obtained by membrane processing where R100 (>100 kDa), R50 (50–100 kDa), R30 (30–50 kDa), R10 (10–30 kDa), R5 (5–10 kDa), and P5 (<5 kDa) after pressurized hot water extraction from the brown seaweed *S. muticum*.

**Figure 5 marinedrugs-22-00016-f005:**
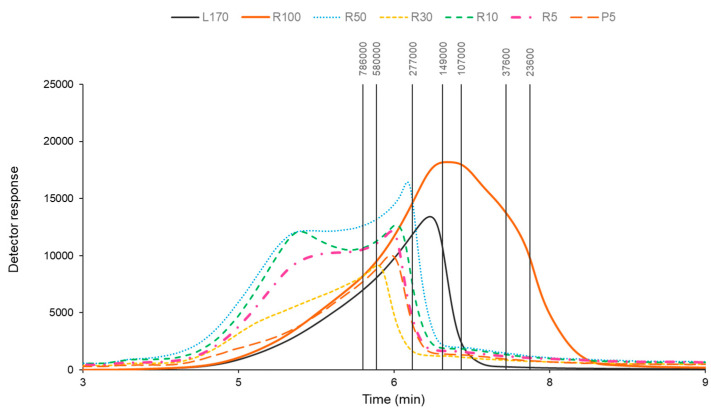
Molar mass distribution profiles of the samples obtained from *S. muticum*: free-alginate liquid extract (L170) and membrane fractions R100 (>100 kDa), R50 (50–100 kDa), R30 (30–50 kDa), R10 (10–30 kDa), R5 (5–10 kDa), and P5 (<5 kDa). Note: vertical lines are the standards in daltons (Da).

**Figure 6 marinedrugs-22-00016-f006:**
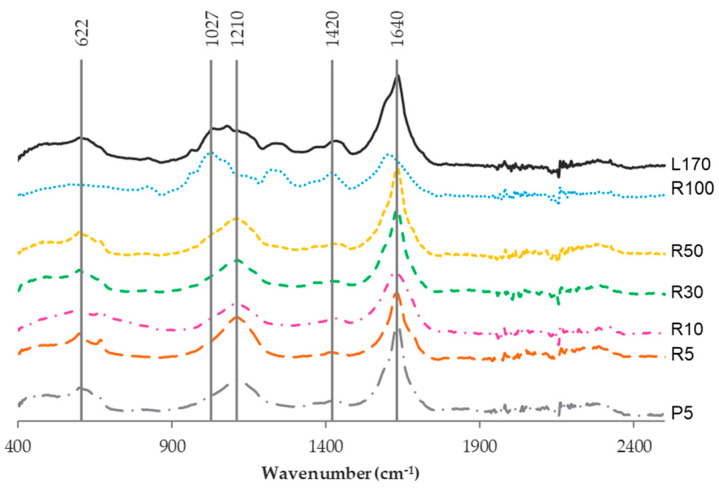
FTIR spectra of the extract obtained from *S. muticum* (170) and the fractions recovered by membrane processing are R100 (>100 kDa), R50 (50–100 kDa), R30 (30–50 kDa), R10 (10–30 kDa), R5 (5–10 kDa), and P5 (<5 kDa).

**Figure 7 marinedrugs-22-00016-f007:**
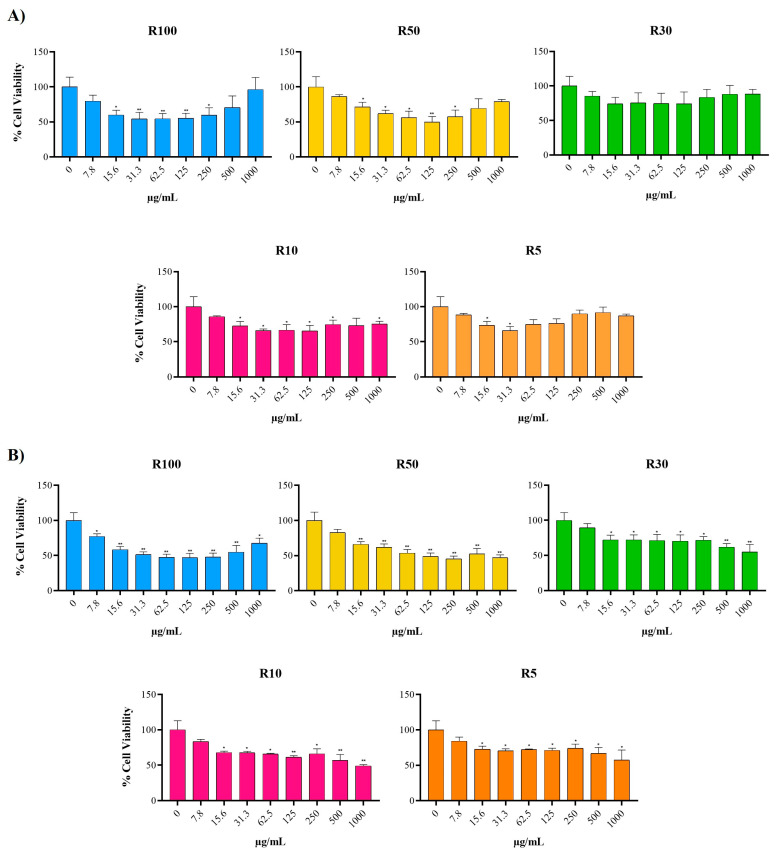
Percentage of cell viability of MCF-7 cells after 48 h (**A**) and 72 h (**B**) of incubation with different concentrations of R100 (>100 kDa), R50 (50–100 kDa), R30 (30–50 kDa), R10 (10–30 kDa), and R5 (5–10 kDa). Statistically significant values are indicated as: * *p* ≤ 0.05, ** *p* ≤ 0.01.

**Figure 8 marinedrugs-22-00016-f008:**
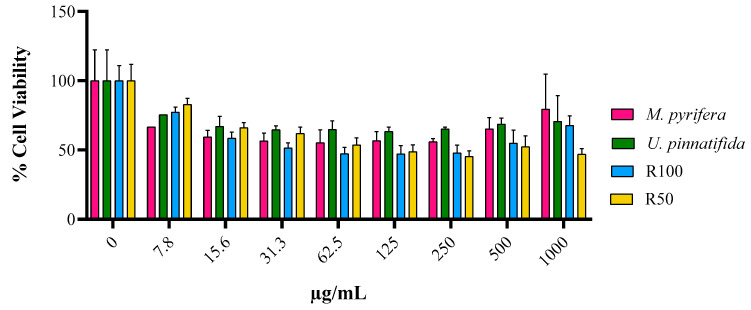
Percentage of cell viability of MCF-7 cells after 72 h of incubation with different concentrations of fucoidans from *M. pyrifera*, *U. pinnatifida*, and the R100 and R50 fractions from *S. muticum*.

**Figure 9 marinedrugs-22-00016-f009:**
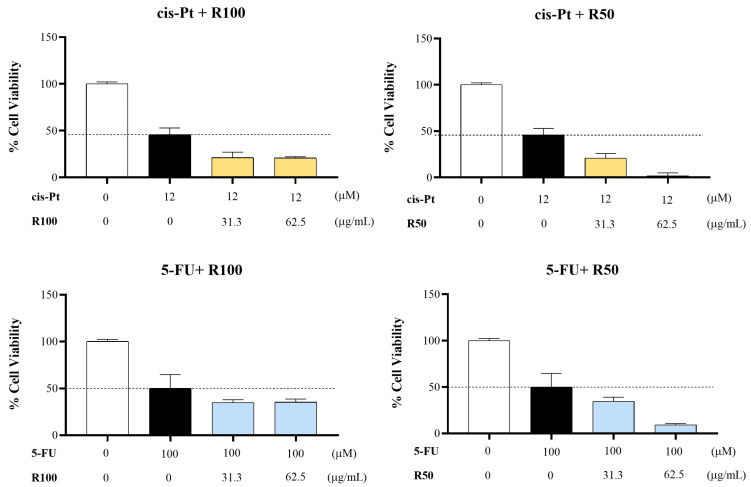
Percentage of cell viability of MCF-7 cells after 72 h of incubation with two different concentrations of R50 and R100 in combination with the IC_50_ of Cis-Pt and 5-FU. IC_50_: half-maximal inhibitory concentration; Cis-Pt: cis-Platinum; 5-FU: fluorouracil.

**Table 1 marinedrugs-22-00016-t001:** Characterization of the initial and pressed raw material and the proximal composition of the solid residue after pressurized hot water extraction at 170 °C.

	Properties	*Sargassum muticum* [15]	*Sargassum muticum* Pressed	Solid Residue
	Moisture (%, d.b. *)	85.59 ± 0.53	67.85 ± 1.23	11.60 ± 0.02
	Ash (%, d.b. *)	17.15 ± 0.07	17.73 ± 0.45	20.88 ± 0.07
	Sulfates (%, d.b. *)	3.46 ± 0.09	0.04 ± 0.00	0.03 ± 0.00
	Proteins (%, d.b. *)	8.47 ± 0.34	8.73 ± 0.67	8.88 ± 0.08
	AIR ** (%, d.b. *)	31.10 ± 0.05	3.83 ± 1.57	24.76 ± 0.19
	Carbohydrates (%, d.b. *)	15.49 ± 0.67	34.85 ± 1.15	24.22 ± 0.43
Minerals (g/kg)	Calcium (Ca^2+^)	18.83 ± 2.23	1.42 ± 0.03	16.22 ± 0.60
	Potasium (K^+^)	23.25 ± 0.35	3.12 ± 0.03	8.60 ± 0.26
	Magnesium (Mg^2+^)	7.50 ± 0.14	5.44 ± 0.06	7.31 ± 0.10
	Sodium (Na^+^)	7.25 ± 0.07	1.26 ± 0.00	43.14 ± 2.21
	Zinc (Zn^2+^)	<10	0.11 ± 0.11	0.04 ± 0.00
Heavy metals (mg/kg)	Copper (Cu^+^)	<7	5.50 ± 0.08	7.36 ± 0.84
	Cadmium (Cd^2+^)	<2	<0.5	0.90 ± 0.07
	Iron (Fe^2+^)	87.33 ± 10.21	265.85 ± 16.98	230.41 ± 10.02
	Lead (Pb^2+^)	<2	0.64 ± 0.02	1.77 ± 0.71

* d.b.: dry bases, *w*/*w*; ** AIR: acid-insoluble residue.

**Table 2 marinedrugs-22-00016-t002:** Characterization of the liquid phases obtained after pressurized hot water extraction and the retentates and permeate obtained by the membrane process: pH, conductivity, sulfate content, protein content in equivalent grams of bovine serum albumin, antiradical properties expressed as TEAC value (trolox equivalent antioxidant capacity), and phloroglucinol content of the diluted hydrolysate (L170) were analyzed.

Fractions	pH	Concentration (g extract/L)	Conductivity (Eq-g CaCl_2_/L)	Phloroglucinol Content (g/100 g)	TEAC Value (g/100 g)	Sulfate Content (g/100 g)	Protein Content (g/100 g)
L170	4.87 ± 0.01	57.20 ± 0.02	54.58 ± 1.62	0.93 ± 0.01	2.35 ± 0.01	2.67 ± 0.09	0.08 ± 0.01
R100	5.28 ± 0.01	5.27 ± 0.01	1.16 ± 0.01	4.18 ± 0.02	10.13 ± 0.01	16.71 ± 1.29	0.53 ± 0.02
R50	6.65 ± 0.01	12.65 ± 0.01	12.78 ± 1.32	0.77 ± 0.01	1.18 ± 0.06	7.52 ± 0.31	0.10 ± 0.01
R30	5.74 ± 0.01	11.62 ± 0.01	14.80 ± 0.31	0.60 ± 0.01	1.43 ± 0.07	6.20 ± 0.02	0.05 ± 0.01
R10	5.77 ± 0.01	12.79 ± 0.01	15.31 ± 0.35	0.67 ± 0.01	2.60 ± 0.49	6.41 ± 0.02	0.07 ± 0.01
R5	5.68 ± 0.01	12.40 ± 0.01	15.16 ± 0.45	0.61 ± 0.01	0.87 ± 0.03	5.60 ± 0.01	0.05 ± 0.01
P5	5.56 ± 0.01	11.55 ± 0.01	15.71 ± 0.26	0.56 ± 0.01	1.24 ± 0.07	4.33 ± 0.02	0.04 ± 0.01

## Data Availability

Data will be available under request.

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
