# Peer review of "Characterization of the Antiproliferative Activity of Sargassum muticum Low and High Molecular Weight Polysaccharide Fractions"

_marinedrugs, 2023, doi:10.3390/md22010016_

Round 1

Reviewer 1 Report

Comments and Suggestions for Authors

This article studied the structural characteristics and functional activities of low and high molecular weight polysaccharides from Sargassum, and conducted in-depth research on the differences in activity caused by molecular weight differences.

1.     Suggest adding the structural characteristics of published algal polysaccharides in the introduction.

2.     Could fractionation obtain relatively uniform polysaccharide components?

3.     Does the relatively low content of carbohydrates have a certain impact on the research results?

4.     Does the difference in molecular weight have a certain impact on the measurement results?

5.     The author should delve into the relationship between structure and function.

6.     Is the time interval between the results of the samples collected in 2016 and published in 2023 too long, and can it affect subsequent repeated experiments?

7.     Suggest adding the following references: Structural characterization and antioxidant activity of a novel high-molecular-weight polysaccharide from Ziziphus Jujuba cv. Muzao. Journal of Food Measurement and Characterization, 2022, 16(3), 2191-2200.

Author Response

After reviewers’ comments and suggestions, we have uploaded a new version of the manuscript, with modifications marked in red so they are easier to distinguish.

We believe the manuscript has considerably improved, especially with a more detailed and richer discussion on the antiproliferative effect of the algae fractions and the influence of the molecular weight.

We are truly grateful for the received input.

REVIEWER 1

This article studied the structural characteristics and functional activities of low and high molecular weight polysaccharides from Sargassum, and conducted in-depth research on the differences in activity caused by molecular weight differences.

1. Suggest adding the structural characteristics of published algal polysaccharides in the introduction.

Thank you for your suggestion. The reviewer is right, the structural characteristics of the other algal polysaccharides was added. Please see between lines 44 - 45.

2. Could fractionation obtain relatively uniform polysaccharide components?

The fractionation carried out in this work was based on membrane cut-off, with different sizes found different molecules but not uniform polysaccharide components because we are working with extracts, a mixture of several compounds.

3. Does the relatively low content of carbohydrates have a certain impact on the research results?

The main bioactive component in brown seaweed is fucoidan, mainly comprised by fucose and sulphate content (with also other carbohydrates). The content of the retentate of 100 kDa and 50 kDa, the more active fractions, represent the higher carbohydrate content. According to the biological properties tested in this work and the results obtained, the authors assume the influence of the carbohydrate content and other bioactive molecules on the results.

4. Does the difference in molecular weight have a certain impact on the measurement results?

At the biological level, we have found that the low molecular weight fractions are more active than the high molecular weight ones, according to previous results in the bibliography. Following the reviewer’s suggestion, we have extended the discussion to include a review on the topic that states clearer the differences in the biological activity between high and low molecular weight fucoidans. Lines 255-269.

5. The author should delve into the relationship between structure and function.

We agree with the relevance of discussing further the physicochemical characteristics and the function of the fucoidan fractions. We have revised the discussion accordingly, comparing also with other results from the bibliography. Please see the revised manuscript, sections 2.5 and 2.6.

6. Is the time interval between the results of the samples collected in 2016 and published in 2023 too long, and can it affect subsequent repeated experiments?

Thank you for your comment, in this case, the samples were collected in 2016 and the experimental was carried out in the following six months, although the manuscript was prepared in 2023.

7. Suggest adding the following references: Structural characterization and antioxidant activity of a novel high-molecular-weight polysaccharide from Ziziphus Jujuba cv. Muzao. Journal of Food Measurement and Characterization, 2022, 16(3), 2191-2200.

We thank the reviewer for the suggestion and we have added other references that were more related with our results, focused on the antiproliferative effect. 

Reviewer 2 Report

Comments and Suggestions for Authors

In this study Diego-González and coworkers tried to analyze the anti-cancer activity of fractions polysaccharides obtained from Sargassum muticum. In the experiments they used many in vitro based methods, including cell-line based in vitro studies. The manuscript may be interesting for the readers, however it needs to be revised. The abstract obligatory needs to be rewritten, since it is very difficult to understand what the Authors meant. I would like to emphasize that I am not going to edit the grammar of the whole manuscript.

Below are my comments:

Authors do not perform any studies showing antitumoral activity – it is effect on MCF7 cells viability (in vitro assay). The title needs to be obligatory rewritten.  

In the introduction should be presented the basic research and the literature basis connected with the goal of the study – the anticancer activity of obtained fractions. There is no explanation why MCF7 cell line was used, as well as why the synergic effect of selected fractions was determined.

Line 274 – present the detailed characteristic of brown seaweed pressed

In all methods present the weight of plant material used for further steps, as well as the volumes of liquids (i.e., 3.3, 3.4, 3.5,3.6,3.7 etc).

In methods used for saccharides concentration there is only volume of sample presented – add their concentrations.

Since Authors mentioned that used MCF7 cell line, why in the methods A549 cell line is presented (MTS)?

In case of fractions, full extract, cis-Pt and 5FU present the stock solution concentrations and the solvent used.

Table 1- what is the unit of data presented as a d.b.?

There are no figure S1, figure S2, figure s3, figure s4 presented.

Part 2.5 - MTS assay is used to determine the cytotoxic activity (the effect on cells metabolism and viability) – showing that the antitumoral effect was demonstrated is inappropriate. Modify this statement within the whole text. This is not the assay for proliferation studies. This part needsto be obligatory rewritten.

Line 196 and methods - Why cells viability was studied after 48 h, not 24?

Figure 6 (line 216) – explain why at 1000 µg/mL cells viability is higher than at 125 µg/mL.

Figure 6 (line 216) – modify the title since the effect of different fractions on cell viability was determined.

Discussion of the results in 2.5 should contain more details (concentration, type of compound, type of cell line, incubation time, etc).

Line 242  - there is another Figure 6. Modify its title in similar manner as for Figure 6 (line 216)

Part 2.6 needs to be rewritten taking into account the MTS assay usage. Explain the rationale for used concentrations of fractions (31 and 62 μg/mL) and cis-Pt and 5FU. Why only fractions R100 and R50 were used?

I suggest to add some other experiments showing the anticancer effect, i.e., apoptosis induction.

The discussion needs to be rewritten taking into account presented comments. The authors should mention the possible way for induction of fractions to the organism to show their potential biological effect. If it is oral, rather other cell lines should be included. If there are breast cancer cells the bioavailability of the fractions should be commented.

In summary, the manuscript requires at the least the major revision.

Author Response

After reviewers’ comments and suggestions, we have uploaded a new version of the manuscript, with modifications marked in red so they are easier to distinguish.

We believe the manuscript has considerably improved, especially with a more detailed and richer discussion on the antiproliferative effect of the algae fractions and the influence of the molecular weight.

We are truly grateful for the received input.

REVIEWER 2

In this study Diego-González and coworkers tried to analyze the anti-cancer activity of fractions polysaccharides obtained from Sargassum muticum. In the experiments they used many in vitro based methods, including cell-line based in vitro studies. The manuscript may be interesting for the readers, however it needs to be revised. The abstract obligatory needs to be rewritten, since it is very difficult to understand what the Authors meant. I would like to emphasize that I am not going to edit the grammar of the whole manuscript.

Below are my comments:

1. Authors do not perform any studies showing antitumoral activity – it is effect on MCF7 cells viability (in vitro assay). The title needs to be obligatory rewritten.  

The reviewer is right about the scope of the viability tests performed. The antitumoral activity is inferred from the capacity of the fractions to decrease the viability of the cell line. However, for clarity and following the reviewer’s suggestion, the word “antitumoral” has been replaced by “antiproliferative” in the title and in the text. Please, see the revised version of the manuscript.

2. In the introduction should be presented the basic research and the literature basis connected with the goal of the study – the anticancer activity of obtained fractions. There is no explanation why MCF7 cell line was used, as well as why the synergic effect of selected fractions was determined.

Following the reviewer’s suggestion, we have included the rationale of using the MCF7 cell line in the introduction. Lines 71-79.

Breast cancer continues to be one of the most prevalent tumors and the first cause of death by cancer in women, despite the latest advances in diagnosis and treatment [16]. Hence, there is an urgent need to find more suitable therapies with low toxicity. The use of natural compounds with antiproliferative activity can be a safe and cost-effective alternative to improve the limited therapeutic effect of conventional therapies, such as the case of chemotherapy.

For that reason, we have also tested the effect of the most active fractions in combination with two of the most frequently used chemotherapeutic drugs (cis-Platinum (cis-Pt) and Fluorouracil (5-FU)).

3. Line 274 – present the detailed characteristic of brown seaweed pressed

In all methods present the weight of plant material used for further steps, as well as the volumes of liquids (i.e., 3.3, 3.4, 3.5,3.6,3.7 etc).

The detailed characteristic of pressed it was added to the sentence, please see on lines 273-274: “S. muticum was pressed (Enerpac RC106, USA), 50 g of wet seaweed applying a force at 2 MPa.”

4. In methods used for saccharides concentration there is only volume of sample presented – add their concentrations.

It was a liquid fraction; the concentration of these fractions was included in Table 2. Please see between lines 160-161. Besides, the Figure 4 represents the results of oligosaccharides in g/100 g extract.

5. Since Authors mentioned that used MCF7 cell line, why in the methods A549 cell line is presented (MTS)?

The A549 (lung epithelial cell line) was used for comparison and to confirm the observations found in the MCF7 cell line. The results with A549 are presented in supplementary files as Figure S2. Please see the supplementary file that was not included in the first round of revision by mistake. We apologise for the mistake and we acknowledge the reviewer’s comment to include it now in the revision process. 

6. In case of fractions, full extract, cis-Pt and 5FU present the stock solution concentrations and the solvent used.

In all cases, milli-Q water was used as a solvent. The stock concentrations were: R100 (11.36 mg/mL), R50 (12.98 mg/mL), R30 (8.68 mg/mL), R10 (12.27 mg/mL), R5 (7.93 mg/mL), cis-Pt (2 mM) and 5FU (40 mM). We have now included this information in the materials and methods section, lines 454-457.

7. Table 1- what is the unit of data presented as a d.b.?

It was amended, please see on Table 1 (lines 117-118), this sentence was inserted: “d.b.: dry bases, w/w”

8. There are no figure S1, figure S2, figure s3, figure s4 presented.

The supplementary material has been added now. Please see also answer to question 5.

9. Part 2.5 - MTS assay is used to determine the cytotoxic activity (the effect on cells metabolism and viability) – showing that the antitumoral effect was demonstrated is inappropriate. Modify this statement within the whole text. This is not the assay for proliferation studies. This part needs to be obligatory rewritten. Discussion of the results in 2.5 should contain more details (concentration, type of compound, type of cell line, incubation time, etc).

We have changed the text and the title of the section accordingly, including those details. Please see the revised manuscript, lines 205-285.

10. Line 196 and methods - Why cells viability was studied after 48 h, not 24?

An initial screening was performed after 24 h of incubation and we found no effects on cell viability. In addition, based on the bibliography and our previous experience with other fucoidans, the effect of these compounds are commonly observed after 48 - 72 h of incubation. In the discussion of the results, we have now included the times used in other works for a better comparison.

11. Figure 6 (line 216) – explain why at 1000 µg/mL cells viability is higher than at 125 µg/mL.

We have included a possible explanation in the discussion, lines 209-229. It has been described before that this artefact could be due to the presence of antioxidant compounds that interfere with the assay (ref: 10.1055/s-2002-32073, 10.3390/antiox8060191; in the manuscript references 31 and 32). The antioxidant compounds can reduce the tetrazolium salt, inducing an apparent increase in cell viability that is most prominent at the highest concentrations tested. For that reason, we have used a second cell viability test that is not dependent on the cell enzymatic activity, such as the xCELLigence system. Please see the supplementary materials and Figure S3.

12. Figure 6 (line 216) – modify the title since the effect of different fractions on cell viability was determined.

The legend of the figure has been changed to:

Percentage of cell viability of MCF-7 cells after 72 h of incubation with different concentrations of fucoidans from M. pyrifera, U. pinnatifida and the R100 and R50 fractions from S. muticum.

13. Line 242  - there is another Figure 6. Modify its title in similar manner as for Figure 6 (line 216)

We thank the reviewer for pointing out this mistake with figure numbering. This has been corrected in the revised manuscript.

14. Part 2.6 needs to be rewritten taking into account the MTS assay usage.

This part has been changed accordingly in the revised manuscript as well as the conclusions.

15. Explain the rationale for used concentrations of fractions (31 and 62 μg/mL) and cis-Pt and 5FU. Why only fractions R100 and R50 were used?

We only used R100 and R50 because these were the fractions that showed the greatest antiproliferative activity in the MCF-7 cell line, as explained in the discussion, lines 291-293.

We selected the lowest concentrations with the highest biological effect that were 31 and 62 μg/mL for the fucoidan fractions. For the chemotherapeutic drugs we selected the IC50 because it is a useful concentration to compare different chemotherapeutic agents. As we explained in the discussion “the use of biological active and safe compounds that can increase the therapeutic effect of a chemotherapeutic drug, while reducing the dose and associated side effects, would be a desired alternative to the available therapies”. For that reason, we tried to use the lowest concentrations with the highest biological effect.

It is worth mentioning that two higher concentrations were also tested for R50 and R100 (125 and 250 μg/mL) but no increase in the antiproliferative effect was observed (data not shown).

16. I suggest to add some other experiments showing the anticancer effect, i.e., apoptosis induction. The discussion needs to be rewritten taking into account presented comments. The authors should mention the possible way for induction of fractions to the organism to show their potential biological effect. If it is oral, rather other cell lines should be included. If there are breast cancer cells the bioavailability of the fractions should be commented.

We agree with the reviewer that adding other experiments to characterize the antitumoral effect of the fractions would be very useful to describe the mechanisms underlying their biological activity. However, the scope of this work was to validate the extraction method to obtain biological active fractions and to compare the activity of the low and high molecular weight compounds. For that reason we have conducted the antiproliferative tests with the fractions alone and combined with the chemotherapeutic agents. The mechanisms underlying the antiproliferative effect of fucoidans has been reviewed in different works (10.1016/j.algal.2019.101748;  10.1016/j.algal.2020.101884; 10.3390/md17010032; 10.1186/s12935-020-01233-8; references in the manuscript: 38, 39, 40 and 41). We have added this information in the revised manuscript, lines 261-269 and 304-311. Regarding the use of fucoidan in a clinical approach to treat breast cancer or other solid tumors, the most common approach in clinical trials is using the fucoidans as a supplement, by doing oral administration (https://clinicaltrials.gov/search?cond=Cancer&intr=fucoidan). However, we think that nanotechnology could be a valuable alternative to target different tumors and release in a controlled manner the fucoidans alone or in combination with other therapeutic drugs as reviewed in https://doi.org/10.1016/j.ijbiomac.2022.07.068.

In summary, the manuscript requires at the least the major revision.

We have performed a major revision and we hope that we have increased the quality of the work accordingly.

Round 2

Reviewer 2 Report

Comments and Suggestions for Authors

I have read the Authors’ comments and improved manuscript- they have answered some of my concerns, however, in my opinion some of the presented results needs to be taken into consideration again.

First of all, the effect on proliferation has not been studied by Authors – with MTT based assay they checked the fractions effect on metabolic activity. Proliferation, therefore the effect on replication and cells division, can be studied with other assays, i.e., BrdU. Therefore, the term “proliferation” in presented manuscript needs to be obligatory removed, and instead can be used terms “cytotoxicity”, “metabolic activity”,” cells viability”.

As “antitumoral” activity  I understand the effect on tumor formation – and Authors did not performed this type of analysis, since their studies are only cell line based in vitro studies. This was my first thought connected with the modification of the title – and the title still needs to be modified.

Still, I do not understand the reason for showing only results for MCF7 cells, and presentation of the effect on A549 as a supplementary material. Please, make the final decision about showing results for only one chosen cell line, and clearly state this in the title of the manuscript.

I have read the explanation about the increase of absorbance when the more concentrated fractions were studied – it is true, but at the same time I suspect that the authors have not performed the blanks containing only medium with the compound, to which the MTS reagent has been added.

In figure S4 it is not known which compound effect is presented on each schemes.

Taking above comments into consideration I suggest the minor revision of the manuscript. Still, I would like to read the improved version of the text. 

Author Response

Please see the authors' cover letter attached.
